# Artificial Intelligence-Based Autonomous UAV Networks: A Survey

**Nurul I. Sarkar** [1],[*] and **Sonia Gul** [2]

1   Computer Science and Software Engineering, Auckland University of Technology, Auckland 1010, New Zealand
2   Network and Communication Department, College of Computer Science and Information Technology, King Faisal University, Al Ahsa 31982, Saudi Arabia
*   Correspondence: nurul.sarkar@aut.ac.nz

**Abstract:** Recent advancements in unmanned aerial vehicles (UAVs) have proven UAVs to be an inevitable part of future networking and communications systems. While many researchers have proposed UAV-assisted solutions for improving traditional network performance by extending coverage and capacity, an in-depth study on aspects of artificial intelligence-based autonomous UAV network design has not been fully explored yet. The objective of this paper is to present a comprehensive survey of AI-based autonomous UAV networks. A careful survey was conducted of more than 100 articles on UAVs focusing on the classification of autonomous features, network resource management and planning, multiple access and routing protocols, and power control and energy efficiency for UAV networks. By reviewing and analyzing the UAV networking literature, it is found that AI-based UAVs are a technologically feasible and economically viable paradigm for cost-effectiveness in the design and deployment of such next-generation autonomous networks. Finally, this paper identifies open research problems in the emerging field of UAV networks. This study is expected to stimulate more research endeavors to build low-cost, energy-efficient, next-generation autonomous UAV networks.

**Keywords:** artificial intelligence; autonomous UAV; AI-based UAV networks; MANET; machine learning; security; privacy

## 1. Introduction

Research on unmanned aerial vehicle (UAV) networks has generated great interest among network researchers recently. For instance, various studies have been carried out focusing on the design and performance evaluation of UAV networks [1,2]. One of the reasons for using UAV networks is to address the challenging terrains, which are very difficult or impossible to be covered by traditional communications networks. UAV networks are excellent at providing optimal performance based on their topology, spectrum efficiency, and context awareness. As human interventions are limited to UAV network operation, network researchers are keen to explore artificial intelligence (AI) techniques to manage these networks effectively. Various applications are being developed using machine learning methods. These applications are designed for both academic research and commercial purposes.

The review question (i.e., objective formulation) we consider in this paper is: *What comprehensive literature survey paper can be formulated on AI-based autonomous UAV networks?* To solve the review question, we have conducted a comprehensive literature survey by selecting more than 100 papers on UAVs focusing on some aspects of autonomous features, network planning, resource management, network access and routing protocols, and energy efficiency. The AI-based UAV networks are discussed next.

*1.1. AI-Based UAV Networks*

Artificial intelligence (AI) is the science used to develop techniques that can think like humans or even beyond humans' intelligence. One of the critical features addressed by AI is the ability to learn and adapt [3]. Therefore, AI techniques are a suitable candidate to apply to UAV networks with their fluid technology and other challenging characteristics. Many network researchers are currently exploring AI applications in UAV network domains. While AI-based UAV network design is an ongoing research area, in this section, we focus on four key areas, including security and privacy, network design, localization and trajectory, and general applications of UAVs that are necessary for the efficient design and deployment of next-generation UAV networks.

1.1.1. Security and Privacy Issues

Security and privacy are always a concern when dealing with a wireless network. This concern even gets more robust when the wireless network we are dealing with has an ever-changing topology. Many studies are being conducted to highlight the security and privacy issues in UAV networks and address these issues using AI techniques. The authors in [4] highlighted various security issues for UAVs with traditional cellular networks. AI-based security solutions are being suggested for various cyber-attacks, including cyber-physical attacks. The authors have proposed using convolutional neural networks (CNNs) and recurrent neural networks (RNNs) to identify and classify high-risk areas and various motion characteristics of UAVs.

An interesting study is presented based on communicating only local information among the neighboring UAVs in a UAV swamp [5]. The simulation results show that a UAV swamp can effectively blanket terrestrial coverage by using local information and voiding various cyber security threats. The simulation is conducted in OPNET Modeler, considering MANET connectivity for all UAVs.

Federated learning (FL) is a new ML technique proposed for distributed Internet of Things (IoT) devices. FL's critical feature is providing a secure communication channel among IoT devices. Reference [6] proposed a UAV-based FL technique based on the 5G heterogeneous networks concepts. Introducing UAVs in the FL enhanced communication efficiency, hence resulting in more reliable and secure network coverage.

1.1.2. UAV Network Design Issues

As UAV networks possess unique characteristics, therefore, the solutions developed to address various needs of other wireless networks such as MANETs and VANETs cannot be used for these networks. Therefore, researchers address the network issues faced by UAV networks a bit differently.

To address the topology control for a swamp of UAVs, an autonomous flock control technique is proposed in [7]. The proposed solution supports the swamp of UAVs during their flight by maintaining their topology in an energy-efficient manner. The authors have used the principles of the Reynolds' Boid model [8], i.e., alignment, separation, and cohesion. The effectiveness of the proposed solution is evaluated by conducting a simulation using the OMNET++ Modeler. In [4], the authors discuss various wireless issues UAV networks are facing and also propose some AI-based solutions for the identified issues. The highlighted issues include high reliability, low latency, efficient handover, and efficient path planning. Various AI techniques are proposed to achieve better network conditions for multiple UAVs.

1.1.3. Localization and Trajectory

Whenever we talk about aerial nodes or vehicles, the challenges of localization and trajectory always come to mind. In a recent study [9], an AI-based trajectory planning for UAV networks was proposed. The main idea is to use a quantum mechanism to support UAVs from the starting point to their destination place. They simulated the system, and

the results obtained from the study [9] show that the proposed solution outperforms the Q-learning technique traditionally used for this purpose.

Localization is a challenging task with unstable network conditions and high mobility of nodes. In [10], an intelligent localization technique is proposed for swarm UAVs. The localization problem is being solved in a 3D model, in which UAV node localization is being realized by limiting the search space in the initial step. This helps reduce the localization error and increases the network convergence speed. The proposed solution is further enhanced by considering the energy-efficient routing for the UAV network. They proposed an algorithm that is evaluated by system simulation. The simulation results of [10] show that localization and coverage time has been improved by using the proposed system.

### 1.1.4. General Applications

Various research studies are being conducted to explore multiple applications of UAV networks. These include but are not limited to using UAV networks as part of cellular networks, vehicular networks, high-risk areas coverage, and effective use of spectrum. In [11], the authors have explored UAV networks for the performance improvement of VANETs. They proposed routing protocols that not only provide better and more reliable coverage for the vehicles involved but also help to identify malicious vehicles. The simulation results presented are promising and show approx. 7% gain in malicious vehicular detection using UAV networks.

### 1.2. Summary of Existing Surveys

With the increase in novel technological solutions being proposed recently in the domain of autonomous UAV networks, surveys of published articles have been conducted to provide blanket coverage to many technologies addressing this research area. In this section, we discuss some relevant latest surveys conducted in this domain.

UAV coverage is one of the vital research areas. Many technologies have been developed to provide better area coverage using UAV networks. A detailed survey [12] is being conducted highlighting various problems from the UAV network's coverage perspective. The authors have classified the problems into groups and considered various constraints. The survey findings highlight the need for more intensive research studies to be carried out exploring the UAV network's coverage, inter-UAV networks' coverage, and end-to-end delays and robustness.

Another survey [13] conducted in 2015 addresses the unique features of the UAV networks, which may not be adopted from the ad hoc mesh networks, vehicular technology, or any other wireless networks that already exist. The UAVs are considered unique based on their dynamic nature, the existence of intermittent links, fluid topology, and energy-conservative constraints. In this survey, the authors highlight issues related to physical, data link, network, and transport layers faced by the UAV networks. Survey findings suggest the need for cross-layer solutions to address the unique needs of the UAV networks. Moreover, the authors encouraged that those solutions that are native to UAV networks should be proposed to address the dynamic and energy-critical needs of the networks.

Another unique feature of the UAV networks has been surveyed by [14]. The authors discussed in detail the channel modeling requirements of the UAV networks and how this impact on network performance and design. An in-depth survey has been conducted exploring the low-altitude platforms with ground infrastructure. Low-cost and low-power solutions are explored with empirical modeling for air-to-air and air-to-ground propagation channels.

The 5G millimeter wave with UAV networks was surveyed in [15]. The authors proposed a novel taxonomy to classify recent work in this area into seven categories from physical layer considerations, i.e., antenna techniques to performance assessments. The survey highlights various open research areas and considerations to make UAV networks a greater success for both industry and research communities. Arafat and Moh have presented the findings of an intensive survey exploring various routing techniques for UAV

networks [16]. The survey classified the routing protocols in various groups with their performance evaluations. The authors have also pointed out various open research areas, including security, performance evaluation, and link performance.

Another interesting study was conducted by Fotouhi and team [17], which highlights various aspects of cellular communications over UAVs. This survey addresses various practical aspects of the transition from a traditional cellular network to a network with flying base stations and relay nodes. The study presented various challenges around standardization and regulations, equipment, and security. The authors also highlighted their lessons learned and future directions to pave a path for upcoming research in the field. While the UAV networks are getting popular for providing better coverage, industry and network researchers have also been actively developing various techniques and solutions to make these newly evolving networks as spectrum efficient as possible. A survey based on the use of a software-defined network (SDN) and network function virtualization (NFV) in UAV networks was conducted in 2020 [18]. This study highlights the interoperability issues in UAV networks and explains how SDN and NFV can be considered a natural fit to solve this issue. The study also brings attention to open research areas of future UAV networks.

A recent study [19] provides an in-depth survey of various applications being developed for UAV networks in current years. While the survey provides information on the recent work in this area, it also provides a detailed taxonomy of multiple UAV networks based on nine classes. Another vital highlight of the paper is proposing the three levels of autonomy based on how multiple UAVs work. Moreover, a detailed discussion of challenges and future trends is also provided. Researchers from both industry and academia have lately been proposing and designing various technologies and solutions for UAV networks. One of the recent survey studies [20] explores UAV networks from both security and privacy perspectives. In this study, the authors discuss issues such as sensor malfunctioning, unreliable communication between aerial and ground stations, and possible leakage of aerial-view images as critical factors in the success of UAV networks. The study suggests borrowing some techniques from MANET and vehicular technologies and molding them to apply to UAV networks to be used to address said concerns.

A survey similar to our work was conducted in 2019 [21]. The survey provides a comprehensive discussion of the various machine learning (ML) techniques being used in UAV networks. The authors provide a classification of ML techniques based on the aspects of networks and communications. The survey proposes the use of ML techniques to achieve optimized performance for UAV networks. The authors proposed using a large amount of multi-source data and applying various deep-learning techniques to achieve a highly effective UAV network.

Another recent study covering the ML techniques to aid the UAV networks is presented in [22]. The adaptive nature of UAV networks along with their context-changing awareness make UAVs a suitable candidate for future communication technologies. With the self-adaptive techniques, machine learning (ML) is regarded as one of the best technologies to aid UAV networks in achieving their communication and reliability goals. Various studies focus on UAV communications perspectives between ground and aerial stations. It was suggested that the use of mobile-edge computing (MEC) in tandem with UAV networks makes it more reliable and effective for future communications.

An interesting study exploring the experimental setup, prototyping, and communication for UAV networks is reported in [23]. In this study, the authors have provided step-by-step techniques for establishing a prototype and experimental setup for UAV networks. This setup includes all stages, i.e., from the selection of aircraft to communications protocols. The authors also explored various protocols that can be used to provide effective and reliable communication in the case of UAVs. Insight into future trends and technologies is also provided, including recommendations for the use of machine learning, 5G cellular UAV networks, and aerial relays. Table 1 provides a summary of related surveys conducted in this area.

**Table 1.** Summary of related surveys.

| Survey Scope | AI-Inspired? | UAV Features Addressed | Limitations | Reference |
|---|---|---|---|---|
| Cooperative UAVs, system deployment | Yes | Coverage, deployment, and nodes used | Obstacles in coverage are not considered | [12] |
| Various UAV networks, routing | Yes | Topology, mobility, reliability, and energy efficiency | System optimization has not been explored | [13] |
| UAV channel modeling, low altitude | Yes | Channel measurement and characteristics, fading | UAVs in dense urban areas are not explored | [14] |
| UAV-assisted and 5G mm wave communications | No | UAV as aerial access, relay, and backhaul | Antenna design, channel modeling, and performance assessment | [15] |
| Routing protocols for UAV networks | No | Topology, position, and cluster-based routings | UAV routing such as link disconnection has not been explored | [10] |
| Integration of UAV and cellular networks | Yes | UAV categorization, standardization, aerial channel modeling, and security | UAV antenna design has not been explored | [17] |
| UAV software-defined network (SDN) and network function virtualization (NFV) | Yes | SDN, NFV, cellular communication, routing, and monitoring | Wireless power transfer has not been addressed | [18] |
| Applications of multiple UAV systems | Yes | Coordination, cooperation, system autonomy | Multiple UAV systems have not been explored | [19] |
| Safety, privacy, and security issues of UAVs | No | Sensor-based attacks, GPS jamming, spoofing, and multi-UAV-based security | UAV privacy and security have not been addressed well | [20] |
| Machine learning for UAV communications | Yes | Channel modeling, positioning, resource management | UAVs for vehicular networks not addressed | [21] |
| UAV-centric machine learning | Yes | Cooperation trajectory planning, channel modeling, mobile-edge computing | Traffic dynamics and channel conditions not explored | [22] |
| UAV prototyping and experiments | No | Cellular UAVs, interference mitigation | Path planning optimization not explored | [23] |
| UAV Channel modeling, link budget | No | Two-ray fast fading, Rician fading, Rayleigh fading | UAV with satellite not explored | [24] |

### 1.3. Main Contribution

The main contributions of this paper are highlighted below.

- We critically review and survey more than 100 published research papers selected from scholarly journals and conference proceedings on UAVs.
- We classify the existing research on UAVs based on their autonomous features. To this end, we focus on the survey of network resource management, multiple access and routing protocols, and power control and energy efficiency of UAV networks. This is a significant piece of work contributing to the design and deployment of the next-generation autonomous UAV systems.
- We identify and discuss areas for open research problems, including UAV network coverage, MAC protocol design, AI algorithm design, and aspects of security, safety, and privacy management.

### 1.4. Paper Organization

The structure and organization of this paper is shown in Figure 1. Section 1 outlines AI-based UAV networks and provides a summary of existing survey of the topic investigated. The main contributions are also highlighted in this section. The background and preliminaries of the proposed survey are provided in Section 2. Autonomous UAV

networks, communication, computation and control, channel modeling, and interference management are discussed. The autonomous features in UAVs are reviewed in Section 3. Resource management, network planning, multiple access and routing protocols, power control, and energy efficiency are reviewed. Section 4 focuses on security, safety, and privacy management. Physical layer security, safety, and privacy are reviewed. The challenges and open research areas are highlighted Section 5, and a brief conclusion in Section 6 ends the paper.

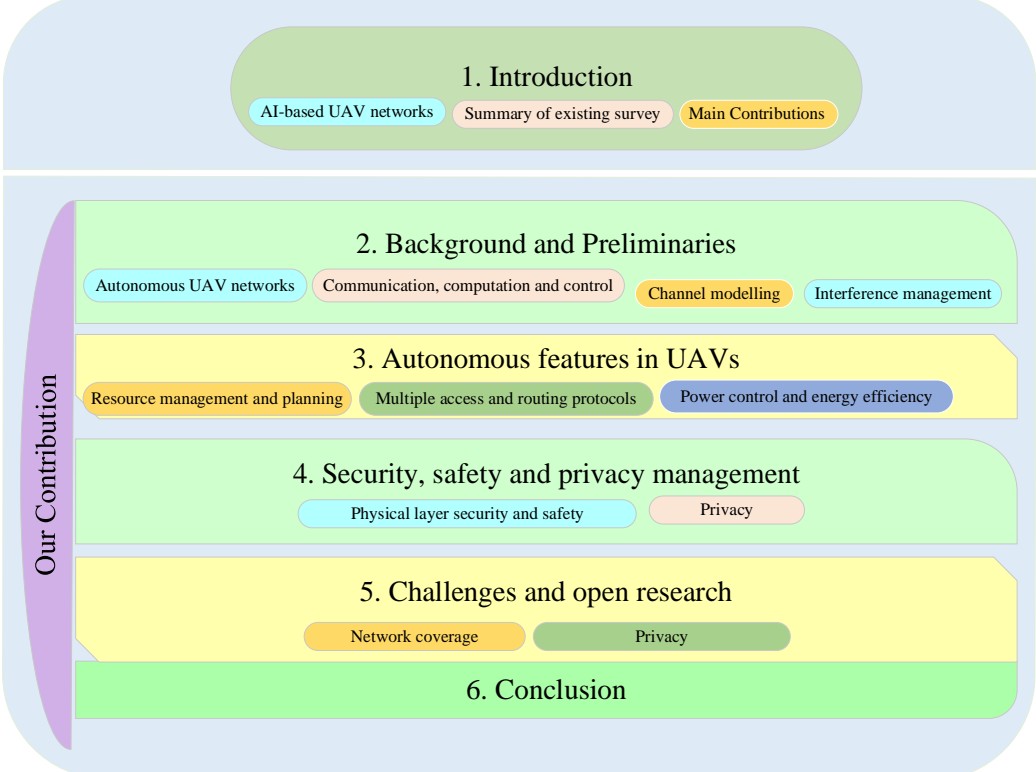

**Figure 1.** Structure and organization of this paper.

## 2. Background and Preliminaries

UAV networks introduce an interesting way to communicate that is different from the traditional practice. In this section, we provide some background and preliminary ideas considered in the early formulization of UAV networks.

### 2.1. Traditional versus Autonomous UAV Networks

Business Insider [25] forecasted that by 2025, the drone industry is expected to hit USD 63.6 billion. The history of UAVs goes back to World War I in 1917I [26]. At those times, the accuracy, privacy, and communication-related issues for UAVs were at high tides, which kept UAVs unreliable for many years. However, as research advances in this field, more reliable solutions for UAV networks are being formulated. However, we restrict our discussion to modern UAV network applications that have been used in recent years.

UAVs are becoming very popular in telecommunications, surveillance, security, and military. The cost efficiency and flexibility in UAV networks have led them to be in the front line of not only communications but also the studies conducted in difficult terrains, e.g., riverbanks, mountain ranges, and forestry.

Aerodynamics plays a vital role in lifting a UAV. Based on the way a UAV is lifted in the air, one can broadly classify these into fixed-wing UAVs (Figure 2a) and multi-rotors UAVs (Figure 2b) [27]. The multi-rotors, also known as rotary wing UAVs, use multiple rotors to generate the uplift using vertical thrust. The number of motors used in these UAVs generally varies from 2 to 8. These UAVs require much energy to support their

vertical take-off and landing mechanisms. On the other hand, fixed-wing UAVs are more energy efficient as they use gliding to save energy. Various studies [28] are being conducted exploring fixed-wing UAVs. In general, fixed-wing UAVs require runways for take-off and landing like traditional planes. In many modern UAV networks, both fixed-wing and multi-rotors UAVs are used.

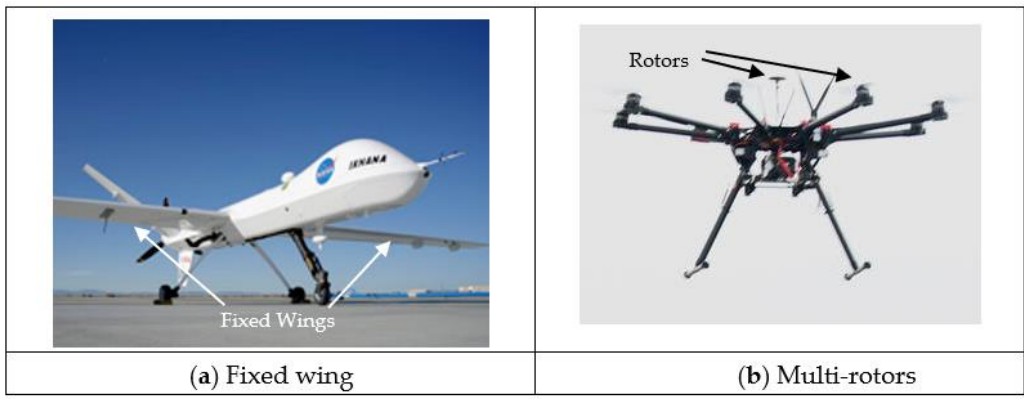

**Figure 2.** Commonly used UAV. (**a**) Fixed-wing UAV [29] and (**b**) multi-rotors UAV [30].

*2.2. Communication, Computation, and Control*

A UAV can be very useful in many areas of daily life. However, to extend the capability of a single UAV, the concept of cooperative UAVs has emerged. Many research studies are conducted to explore the possibilities a UAV network can provide.

In [31], a decentralized model for predictive control is being proposed. The model is being demonstrated on a group of UAVs. The simulation results show that the proposed model is computationally efficient as compared to central approaches.

Another interesting study is conducted by Bellingham, J. et al. [32]. In this study, a control system is developed to optimize the coordination among multiple UAVs. The authors address key issues such as trajectory optimization, resource allocation, and goal assignment. The proposed solution combines the trajectory and resources to assist with the overall control of the UAV fleet, which includes cooperative path planning and multitask allocations.

Whenever multiple things are working together, conflicts may arise. A study exploring conflict resolution in a UAV fleet is presented in [33]. A model to identify the conflicts and then resolve them is proposed in this paper. The authors define the flight plan by using the intermediate waypoints and maintaining the UAV's velocities. The simulation and experiment results show that the proposed system is highly scalable and quickly executable. Another study proposed the use of a thermal lift to improve the flight endurance of cooperative UAVs [34]. The simulation results suggest that there is a significant chance of improvement in finding the lift with cooperative UAVs.

*2.3. Channel Modeling*

Channel modeling is one of the crucial parts of UAV networks. In the coming days, we are expected to see UAVs as a regular entity within our aerospace domain. Therefore, discussions on the use of channels by these UAVs become vital.

A comprehensive survey on UAV channel modeling is conducted in 2019 [24]. In this survey, authors have categorized the UAV channel usage into three, i.e., air to ground, air to air, and ground to ground. The authors also discussed the link budget and channel fading characteristics.

Another detailed survey exploring the temporal and spatial characteristics of non-stationary channels is performed in [14]. In this survey, authors explore the areas that require much attention from the research community, e.g., exploring airframe shadowing. The study also provides statistical channel modeling to support successful UAV communications.

To enhance the performance of UAV communications in shadowing scenarios, a new channel model is proposed in [35]. The idea is to provide a generic channel model with a less complex channel selection. The proposed model and techniques are also verified using empirical data.

### 2.4. Interference Management

Interference management is one of the key considerations of any wireless communications. In UAV communications, interference also plays a big role in determining the system performance. Many network researchers have explored the effect of interference on UAV network performance. Most recent research studied interference management for UAV networks either by considering UAVs as the extension of cellular networks or by pairing them with power control. In [36], a path planning protocol focusing on latency and interference for ground networks was studied for UAV networks. In this paper, the interference mitigation technique based on game theory is investigated, and a learning algorithm using the deep reinforcement approach is proposed. Each UAV acts as a player and decides on its path while minimizing the latency and interference for the ground network. Another similar study was conducted in [37]. This study also utilizes game theory, but this time, the goal is to maximize the energy efficiency of the UAVs while minimizing the latency and interference for the ground network. A study based on the approaches of using UAVs with cellular networks is investigated in [38]. In this study, the UAVs are used to assist in the integrated access and backhaul (IAB) cellular in-band networks and reduce interference at various levels of networks. The problem is formulated as a maximization problem for the network sum targeting various network factors and features. The authors claim that the results achieved in this study outperform many existing techniques.

When it comes to interference management, power control is another feature that cannot be ignored. To lower the interference, a machine-learning approach is proposed in [39]. In this approach, the idea is to manage the overall network interference by controlling the UAVs/drone power and location. The proposed framework uses K-means and affinity propagation. The simulation results and discussions provided in the paper are promising for managing network interference. Another study conducted by Zhang, S. et al. [40] addresses the same problem but a bit differently. Here, the authors have tried to manage the interference by dividing the problem into two sub-problems, i.e., finding the optimal transmit power for a given UAV trajectory and then also considering the best trajectory for the fixed transmit power. This helps in finding the optimum solution for the overall network. Zhang, J. et al. [41] address the same problem in a bit different way. The authors focus on not only power control to minimize the interference but also consider UAV clustering to address the problem. The solution is designed as a game to maximize the sum rate of the whole system.

### 3. Autonomous Features in UAV Networks

To enhance the overall performance of the UAV networks and to address some specific problems, new features in the network are being designed as autonomous features. This approach not only provides optimum solutions for the targeted problems but also supports the dynamic properties of a UAV network. Following is a summary of the research community's efforts in designing autonomous features for UAV networks. Figure 3 illustrates the three categories of autonomous features in UAVs. These features of classification include (i) resource management and network planning; (ii) multiple access and routing protocols; and (iii) power control and energy efficiency.

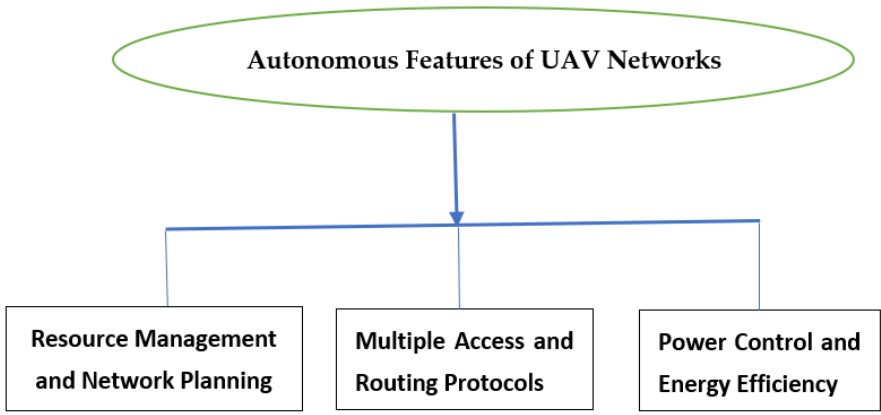

**Figure 3.** Autonomous features classification for UAV networks.

*3.1. Resource Management and Network Planning*

Optimal management and planning of network resources are very crucial for the success of any network. This is particularly true when it comes to networks where human intervention is minimal. For UAVs, this feature is being focused on by many researchers, and various challenges are being identified and addressed. Table 2 provides a summary of related surveys on UAV network resource management and planning.

**Table 2.** Summary of related survey: UAV network resource management and planning.

| Scope | Autonomous Features | Computational Intelligence | Channel Modeling | Interference Management | Security and Safety | Reference |
|---|---|---|---|---|---|---|
| Cooperative path planning, collisions | Path planning | √ | Non-convex modeling | - | - | [31] |
| Cooperative path planning, resource allocation | Path planning | √ | - | - | - | [32] |
| Cooperative UAVs, trajectory conflicts | Conflict detection and resolution | √ | - | - | - | [33] |
| Channel modeling | | √ | Shadowing channel | √ | - | [35] |
| Interference management | Path planning | √ | Rician distribution | √ | - | [36] |
| Interference-aware path planning | Path planning | √ | Free-space path loss with 6 GHz | √ | - | [36] |
| Interference management | Spatial configuration | √ | Customized | √ | - | [38] |
| Interference management | Transmission power, trajectory planning | √ | Large-scale path loss | √ | - | [40] |
| Interference management | Transmission power | √ | Free-space path loss | √ | - | [41] |
| Resource management | Energy consumption, transmission power | √ | Various | √ | - | [42] |
| Collision free navigation | Trajectory planning | √ | - | - | - | [43] |
| Risk-aware path planning | Path planning | √ | - | - | - | [44] |

**Table 2.** *Cont.*

| Scope | Autonomous Features | Computational Intelligence | Channel Modeling | Interference Management | Security and Safety | Reference |
|---|---|---|---|---|---|---|
| Mobility challenges | | - | - | - | √ | [45] |
| Resource management | User association | √ | Free-space path loss | √ | - | [46] |
| Delay-aware throughput maximization | Trajectory planning | √ | Free-space path loss | - | - | [47] |
| UAV placement | Energy efficiency and optimization | √ | Path loss outdoor/indoor penetration | - | - | [48] |
| Collision free navigation | Trajectory planning | √ | - | √ | - | [49] |
| Swarm-based UAV | Path planning | √ | - | - | - | [50] |
| Physical layer security | security and cooperation | √ | Free-space path loss | √ | √ | [51] |
| Secure UAV communication | Cooperative scheduling | √ | Free-space path loss | - | √ | [52] |
| Physical layer security | Cooperative trajectory and optimization | √ | Free-space path loss | √ | √ | [53] |
| Physical layer security | Cooperative resource allocation | √ | Free-space path loss | - | √ | [54] |
| Quality of Experience (QoE) | Cooperative resource allocation | √ | LOS and Non-LOS | - | - | [55] |
| Secure UAV communication | UAV defense | √ | - | - | √ | [56] |
| Software-defined radio (SRD) | Localization of unwelcomed UAVs | √ | - | √ | √ | [57] |

In [42], the autonomous distribution of resource management for UAV networks is being discussed. The authors have used the concept of game theory and highlighted various game models that may be used for optimal resource management among UAVs. In specific, five-game theory models were discussed, including coalition, potential, graphical, mean-field, and Stackelberg. Each model is explained based on its goals, design of utility function, and strategies, which implies their application areas.

Reference [43] focused on real-time planning for UAV's path under dynamic conditions. A discrete algorithm, along with a probabilistic graph, is being used to achieve a path with no collisions. Another similar study to explore path planning for UAVs is conducted in [44]. In this study, the information from both static and dynamic paths is utilized to provide a path. A step-by-step adaptive path planning technique is proposed to achieve optimal results. Another interesting idea is that UAVs are considered additional users in the 5G cellular networks [45]. The study highlights various challenges that are encountered by service providers to facilitate new users.

### 3.2. Multiple Access and Routing Protocols

Multiple access and routing protocols are another challenging domain for UAV networks. As UAVs are supposed to be co-located with other networks, the natural choice for multiple access for UAVs was initially space division multiple access techniques [58–60]. However, as time passes, many researchers have started exploring other techniques that prove to be more efficient in the case of UAV networks. Most of these include orthogonal, non-orthogonal, or rate-splitting techniques. Table 3 provides a summary of related surveys on UAV multiple access and routing protocols.

In orthogonal multiple access techniques, the interference is reduced by ensuring that the simultaneous communications are orthogonal to one another. Multiple research studies [46,61] have been conducted to achieve this goal. Some propose to use time division multiple access (TDMA) and frequency division multiple access (FDMA) in their system models. Another group of researchers focused on using code division multiple access (CDMA) [62–64]. On the other hand, orthogonal frequency division multiple access (OFDMA) is explored by some researchers [47,65,66], while some other research studies are conducted exploring space division multiple access (SDMA) [67–72]. As we can only have a limited number of orthogonal communication channels, spectrum efficiency is reduced using this approach.

To overcome the above-stated limitation of orthogonal multiple access techniques, much work is being performed using non-orthogonal techniques, mainly using joint optimization of the base stations (BSs), including power [73–75], trajectory [76–78], and placement [79,80]. Although non-orthogonal multiple access techniques can perform better by covering many users with non-orthogonal yet correlated channels, however, there is a case when its performance can degrade exponentially, i.e., when the number of antennas is greater than the number of users scheduled. To avoid such circumstances, another effective technique is explored.

Rate-splitting multiple access techniques for UAVs seem to overcome the problems of both orthogonal and non-orthogonal multiple access. Hence, many research studies have been conducted to explore what we can achieve using this technique. Most of the work in this area is based on the concept of the proposal of a joint rate among multiple UAVs to achieve the most optimal network solutions [81–86].

### 3.3. Power Control and Energy Efficiency

For drones and unmanned aerial vehicles (UAVs), power control and energy efficiency are other very critical realms. As these network nodes are not on the ground, an uninterrupted supply of power and efficient utilization of energy become very crucial. Many research studies have addressed this area and proposed several solutions, some targeting multiple power sources, including battery [87,88], hydrogen fuel [89,90], solar [91,92], and hybrid [93,94], that can be utilized by the UAVs. While others focus on how efficiently energy can be consumed. Energy efficiency affects all the operations of a UAV. Many studies have been conducted to explore UAV placement for optimal energy efficiency [48,95].

Another aspect explored in this domain is path planning. Various approaches are being used to explore this area, including sample-based [49,96], space search [50], and biological searching techniques [97–99].

**Table 3.** Summary of related survey: UAV multiple access and routing protocols.

| Scope | Autonomous Features | Computational Intelligence | Channel Modeling | Interference Management | Security and Safety | Reference |
|---|---|---|---|---|---|---|
| Channel access | Cyclic multiple channel access | √ | Free-space path loss with LOS | - | - | [61] |
| MAC protocol | Energy consumption, Packet-error-rate (PER) | √ | Free-space path loss with LOS | - | - | [62,63] |
| Performance evaluation of MAC | PER | √ | Rician fading | - | - | [64] |
| Trajectory optimization | Trajectory planning | √ | Free-space path loss with LOS, correlated Rician fading, Rayleigh fading, Rician K-factor | √ | - | [67] |

**Table 3.** *Cont.*

| Scope | Autonomous Features | Computational Intelligence | Channel Modeling | Interference Management | Security and Safety | Reference |
|---|---|---|---|---|---|---|
| mm wave UAV cellular network | Beam forming | √ | Quasi-static, Rayleigh fading | √ | - | [68] |
| Channel access with time-modulated array (TDM) | Beam forming, performace | √ | Free-space path loss with LOS | √ | - | [70] |
| Trajectory optimization | Trajectory planning, power control | √ | Additive white Gaussian noise (AWGN) | √ | - | [74] |
| MAC protocol | Power optimization | √ | Rician fading | √ | - | [75] |
| MAC protocol | Throughput optimization | √ | Free-space path loss | - | - | [76] |
| MAC protocol | Power optimization | √ | Free-space path loss with LOS | √ | - | [77] |
| MAC protocol | Trajectory planning, resource management | √ | LOS and Non-LOS (NLOS) | √ | - | [78] |
| MAC protocol | Power optimization | √ | LOS, NLOS | √ | - | [79] |
| MAC protocol | Power and placement optimization | √ | Additive white Gaussian noise (AWGN) | √ | - | [80] |
| Rate-splitting | Beam forming | √ | Additive white Gaussian noise (AWGN) | √ | - | [82] |
| Rate-splitting | Spectral efficiency | √ | Additive white Gaussian noise (AWGN) | √ | - | [86] |

## 4. Security, Safety, and Privacy Management

As technology evolves and expands, its security is compromised. Security, safety, and privacy are very crucial domains that need to be handled effectively and efficiently. We have categorized this area into the following two groups (as shown in Figure 4).

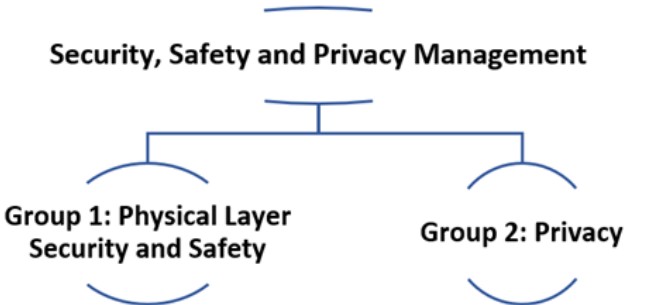

**Figure 4.** Security, safety, and privacy classification of UAV networks.

### 4.1. Physical Layer Security and Safety

At a broader level, we may define physical layer security as techniques to secure our communication over the channel against various attacks, including eavesdropping or jamming. Two typical UAV communication scenarios are illustrated in Figure 5.

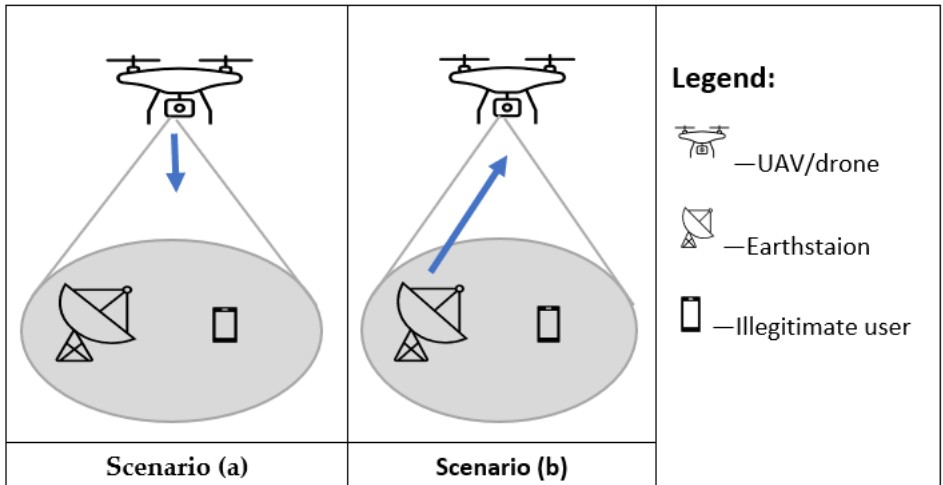

**Figure 5.** UAV communication scenarios: (**a**) UAV communicating to a ground station, (**b**) ground station communicating to UAV.

Scenario (a) in Figure 5a illustrates the situation when the UAV is trying to communicate with the Earth station. In this case, it is much easier for any illegitimate user to eavesdrop on ongoing communication. However, if the Earth station tends to communicate with UAV, eavesdropping is not that simple. Scenario (b) (Figure 5b) has a natural advantage over Scenario (a).

One of the solutions to the above-stated problem is identified as tracking or localizing the potential eavesdropper. Many research studies [100–102] have focused on this and proposed solutions, including mounting cameras or radars to identify or track any potential attacker. Introducing an anti-jamming signal or noise is another solution proposed by researchers [51,103].

The two scenarios illustrated in Figure 5 are commonly classified as passive eavesdropping attacks. However, active eavesdropping is more harmful to the network compared to passive eavesdropping. In active eavesdropping, the attacker tends to attack and play with the main channel of communication. Various solutions are proposed to avoid active eavesdropping attacks as well. One of the popular solutions is by using joint-trajectory and resource allocation for the UAVs [52–54]. This solution suggests a careful enhancement to the joint-trajectory design to reduce the signal strength to the potential illegitimate links while providing a strong signal to the main and other targeted nodes. Another solution proposed by researchers [55,76] is to have a limited number of onboard resources. This will help in designing careful solutions for the UAVs around the ground node and hence reduce the chances of unwanted access. Table 4 provides a summary of related surveys on UAV security, safety, and privacy management.

**Table 4.** Summary of related survey—UAV security, safety, and privacy management.

| Scope | Autonomous Features | Computational Intelligence | Channel Modeling | Interference Management | Security and Safety | Reference |
|---|---|---|---|---|---|---|
| Resource management | Energy consumption, trajectory planning | √ | Free-space path loss with LOS | - | - | [65] |
| Computation optimization with energy management | Computation performance, energy consumption | √ | Block-fading, LOS | √ | - | [66] |
| Electric UAV, Fuzzy state machine | Energy management | √ | - | - | - | [87] |

**Table 4.** *Cont.*

| Scope | Autonomous Features | Computational Intelligence | Channel Modeling | Interference Management | Security and Safety | Reference |
|---|---|---|---|---|---|---|
| Compressed hydrogen, fuel cells | Energy management | √ | - | - | - | [89] |
| Hydrogen, fuel cells | Energy management | √ | - | - | - | [90,93] |
| Solar power | Energy management | √ | - | - | - | [91,92] |
| hybrid fuel | Energy management | √ | - | - | - | [94] |
| UAV backhaul network | Energy efficiency, placement optimization | √ | Free-space optical link (FSO), LOS | √ | - | [95] |
| Genetic algorithm | Energy efficiency | √ | - | - | - | [98] |
| Genetic algorithm | Cooperative path optimization | √ | - | - | - | [99] |
| Physical layer security | Jamming power | √ | Rayleigh fading channel | √ | √ | [102] |
| Secure UAV communication | Jamming power | √ | Free-space pathloss with LOS | - | √ | [103] |
| Co-channel interference management | Transmission power | √ | Free-space path loss with line of sight (LOS) | √ | - | [39] |

Another related area explored by some researchers is how you can defend your network from illegitimate UAVs/drones. This seems to have occurred when other UAVs may be trespassing on your network and accidentally receiving some of your communication or when an eavesdropping attack is being launched by a UAV. In either case, that UAV is not part of our network and should not be able to intercept our communication. To address this problem, many research studies proposed reinforcement-based learning [56] and various detection mechanisms to identify unwanted UAVs in the network [57].

*4.2. Privacy*

Privacy is vital to our society and community. Where the advancements in technology such as UAVs and drones facilitate us in many ways, the threat to people's privacy also rises. Many studies [104,105] are conducted to explore people's views about drones and how they feel these are affecting their privacy.

In [106], an attempt is made to clear up the name of UAVs and drones in terms of invading privacy. Authors claimed that drones are not using any new technology; rather, they are using a mix of existing technologies that have already been approved concerning privacy. The study suggested looking at UAVs and drones as aircraft, but because they are unmanned, they need the camera to perform their operations. Moreover, the article also classifies UAVs and drones as major data collection nodes, and the quality of the images is not so high as to invade people's privacy.

**5. Challenges and Open Research Areas**

Unmanned aerial vehicles (UAVs) are an emerging research area. While various research works have been reported in the networking literature, the system performance issues and challenges have not been fully solved yet. The challenges and open research areas for contribution are highlighted below.

*5.1. Network Coverage*

Many studies have been conducted to improve the network coverage for UAVs. However, a significant reduction in network coverage can be seen when UAVs and drones

are moving at a low speed [107]. This can be considered an open research area where the network coverage can be increased by introducing some changes to the existing mobility models.

Furthermore, to provide network coverage to disaster-affected areas right after earthquakes and tsunamis, potentially emerging technologies such as 5G and UAVs can be used. Certainly, this is another open research area where network coverage can be provided in the affected areas using UAV-mounted small cellular base stations.

### 5.2. MAC Protocol Design

The use of the right MAC protocol is very important in achieving high efficiency for the UAV-incorporated networks. Many network researchers have proposed various techniques to improve the traditional MAC protocols to make them suitable for UAVs. However, to achieve high-efficiency requirements for the UAV networks, the MAC protocols should be designed specifically for UAVs. The cognitive radio approach can be used as a potential partner in designing MAC protocols for UAVs-assisted networks.

### 5.3. AI Algorithm Design

As most of the UAV's functions are performed without direct intervention from humans, AI is inevitable in UAV-assisted networks. Various deep-learning techniques can be applied to some specific application areas for UAVs to achieve optimal system performance. Such areas include but are not limited to emergency response, event coverage, and rural community services.

### 5.4. Privacy and Security

As discussed in Section 4 Scenario (b), many researchers have privacy concerns about UAVs and drones, especially when there are cameras involved. A research gap can be bridged here by introducing some other technologies to UAVs and drones, such as LiDAR and other sensors to move around. A recent study [108] has emerged targeting this but exploring from a privacy perspective may provide an additional edge. Moreover, to ensure secure communication through UAVs, blockchain technology can also be used in tandem with other security features.

The main challenge is to implement data security in flying cellular base stations. To tackle the concerns about data security, the blockchain-based framework can be used to provide a security layer in the communication between (1) drone to drone and (2) user to the drone. Certainly, this is an important open research area that can be investigated for secure UAV systems.

## 6. Conclusions

A comprehensive survey on AI-based autonomous UAV networks is presented in this paper. The survey focused on key aspects of UAV autonomous features, network resource management, channel access, routing protocols, security, and privacy management. Research findings show that AI-based autonomous UAV networks are a technologically viable paradigm for providing a cost-effective solution in the design and deployment of next-generation networks. A concerted effort among industry and academia and close cooperation with various government agencies, including telecommunication sectors and regulatory bodies, is required for its success.

Finally, we identified and discussed promising future research areas, including network coverage, access protocols, AI algorithms, and the security and privacy of UAV networks. New research programs are required to create efficient UAV network architecture and protocols for addressing issues and design challenges of AI-inspired autonomous UAV communication networks. The integration of AI algorithms for a sub-optimal low-complexity solution is suggested for future work.

**Author Contributions:** Conceptualization, N.I.S.; methodology, S.G. and N.I.S.; software, N.I.S.; validation, N.I.S. and S.G.; formal analysis, S.G. and N.I.S.; investigation, S.G. and N.I.S.; resources, N.I.S.; data curation, S.G. and N.I.S.; writing—original draft preparation, S.G.; writing—review and editing, N.I.S.; visualization, S.G. and N.I.S.; supervision, N.I.S.; project administration, N.I.S. All authors have read and agreed to the published version of the manuscript.

**Funding:** This research received no external funding.

**Institutional Review Board Statement:** Not applicable.

**Informed Consent Statement:** Not applicable.

**Data Availability Statement:** Not applicable.

**Acknowledgments:** We thank Minh Nguyen (Head, Computer Science and Software Engineering, Auckland University of Technology) for providing Research Assistance support.

**Conflicts of Interest:** The authors declare no conflict of interest.

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
