# Peer review of "Artificial Intelligence-Based Autonomous UAV Networks: A Survey"

_drones, doi:10.3390/drones7050322_

Round 1

Reviewer 1 Report (Previous Reviewer 1)

Reviewer’s Comments

The manuscript titled, “Artificial Intelligence-Based Autonomous UAV Networks: A Survey”, with the Manuscript id: Drones-2409288, is updated as per the provided comments. I recommend that the manuscript be accepted.

I wish the authors great success,

Reviewer 2 Report (Previous Reviewer 3)

Dear authors! Thank you for carefully following the suggestions. While not all of them are satisfied, I found the work on the material significant, and the paper is notably improved. So, the work can be published.

English language is fine

This manuscript is a resubmission of an earlier submission. The following is a list of the peer review reports and author responses from that submission.

Round 1

Reviewer 1 Report

eviewers Comments

Major revision is being suggested for the manuscript id: drones-2307656, titled, “Artificial Intelligence-Inspired Autonomous UAV Networks: A Survey”. The following are specific comments; the author must revise the manuscript and prepare a rebuttal to the comments for further review.

1.      Author must avoid the use of first person and third person in the research writing.

2.      As authors mentioned this is a survey paper but author have not cited any paper/article till line no 55.

3.      Line no 90-91: “The simulation………..this purpose”, did authors simulated?

4.      Line no 98-99: “The simulation………proposed algorithm”, did authors of this article proposed an algorithms and simulated?

5.      Line 159-154: “Lession…….said field”, what is the significance of such statements? Kindly check the whole manuscript and update such vauge sentences.

6.      Line no 161-162: “Several……….UAV networks’, is it allowed to use several, higher lower and many without the backing support of the numbers in research writing? Kindly check the whole manuscript and correct.

7.      The flow of the introduction is dangling and not connected. Authors have redrafted the other authors work (1 article/paragraph), and not focused on the flow. Kindly check it and club 2-3 paragraphs in 1 and only write the major takeaways of the cited articles.

8.      The research gap formulation is very poor and no proper review questions (Objectives formulated).

9.      Line no 233-234: “The business…….billion.” what is the significance of such statements in the body of the text? It must be in the first or second sentence of the paragraph to broadly provide the global scenario of the domain.

10.  Line 298-299, 314-315: check the sentence, and rewrite.

11.  Line 357-358: who is new user?

12.  As the title of the presented work is, “Artificial Intelligence-Inspired Autonomous UAV Networks: A Survey”, but here authors have not discussed anything about the artificial intelligence and associated technicalities for the UAV networks. It is suggested to revise the entire manuscript and properly redraft the manuscript to satisfy at least the title’s requirements.

13.  Section 6 must be conclusions instead of concluding remarks.

14.  Rewrite the conclusions not more than 150 words. Author must convey only the important takeaways from the work. In addition, if the author like, he or she can include 1-2 sentences outlining the next directions of research.

I wish authors a great success.

Reviewer 2 Report

Authors discussed the following points:

1. Authors critically reviewed and surveyed more than 100 published research papers selected from scholarly journals and conference proceeding on UAVs. 

2. Authors classified the existing research on UAVs based on the autonomous features.

3. Authors focused on the survey of network resource management, multiple access and routing protocols, and power control and energy efficiency of UAV networks.

4. Authors claimed that it is a significant piece of work contributing to the design and deployment of the next generation autonomous UAV systems. 

5. Authors identified and discussed promising future research areas including UAV network coverage, MAC protocol design, AI algorithm design, and aspects of security, safety, and privacy management.

6. Presented very well and references are cited properly

Reviewer 3 Report

Dear authors! The paper considers a cutting edge technology of drone networks, which is a next step of technical evolution of UAVs.

While a large amount of literature sources is analyzed and some conclusions are made, the main idea of the paper is a bit blurred, and some important details are missed.

1. Please clarify what does it mean "AI-inspired"? The term "AI-based" is much more clear, but is seldom used in the text. I propose adding a definition of  "AI-inspired" system or using another term. Also, there is a lack of information on AI algorithms used in drone networks, so the title of the paper does not exactly match its content. Please add one section on AI-based solutions or change the focus of the paper.

2. Subsection "Paper organization" is redundant. And the illustration of it in Fig. 2 looks awkward, as if there was an extra space in the paper and nothing was to add. Please exclude it from the text.

3. Illustrations in Fig. 3, Fig. 5 and Fig. 6 are non-informative. Please add more details otherwise they look trivial and should be excluded.

4. Some references are not relevant. For example, ref. [27] should imply classification of drones. But, it is dedicated to a short NASA press release on Traffic Management technologies. Please check references again much more carefully.

My overall impression is as follows. From the text it is unclear what is this paper exactly about. It looks more like a hastily compiled essay rather than a strict theoretical work. Despite a large amount of interesting details and good expertise presented in the work, I propose careful rewriting of the manuscript with clear focus and correctly referenced papers.